# Assessing Commensality in Research

**DOI:** 10.3390/ijerph18052632

**Published:** 2021-03-05

**Authors:** Henrik Scander, Agneta Yngve, Maria Lennernäs Wiklund

**Affiliations:** 1School of Hospitality, Culinary Arts and Meal Science, Örebro University, 712 02 Grythyttan, Sweden; 2Department of Nutrition, Dietetics and Food Studies, Uppsala University, 751 22 Uppsala, Sweden; agneta.yngve@ikv.uu.se; 3School of Health Sciences, Örebro University, 702 81 Örebro, Sweden; 4Department of Public Health and Sports Science, University of Gävle, 801 76 Gävle, Sweden; maria.lennernas@hig.se

**Keywords:** eating together, conviviality, gastronomy, meal, food studies, dining, eating practice, multidisciplinary

## Abstract

This scoping review focuses on the assessment of commensality in research and attempts to identify used methods for performing research on commensality. It reflects a multidisciplinary research field and draws on findings from Web of Science Core Collection, up to April 2019. The empirical material consisted of 61 studies, whereof most were qualitative research, and some were of quantitative character, including very few dietary surveys. The findings show nine papers categorized as using quantitative approaches, 52 papers were categorized as qualitative. The results show a wide variety of different ways to try to find and understand how commensality can be understood and identified. There seems to be a shift in the very concept of commensality as well as some variations around the concept. This paper argues the need to further investigate the importance of commensality for health and wellbeing, as well as the need to gather data on health and health-related behaviors, living conditions and sociodemographic data in parallel. The review shows the broad-ranging areas where commensality is researched, from cultural and historical areas to ethnographic or anthropological areas over to dietary assessment. To complement large dietary surveys with methods of assessing who you are eating with in what environment should be a simple way to further our knowledge on the circumstances of meal intake and the importance of commensality. To add 24-h dietary recall to any study of commensality is another way of identifying the importance of commensality for dietary quality. The use of mixed methods research was encouraged by several authors as a good way forward in the assessment of commensality and its importance.

## 1. Introduction

Researchers that find an interest in commensality research will discover that there is no easily identified common research methodology. A number of studies dealing with family meals or social eating, studied as a background for this scoping review revealed that the overlap of studies using the search term “family meal” or “eating together” with the term “commensality” was often inclusive. A recent meta-analysis of family meals and health showed that there is convincing evidence showing that family meals have an important impact on the future meal frequency and nutritional health of children [1]. The definitions of “family meals” has, however, been shown to be inconsistent and concentrating on number of family meals per week [2]. Furthermore, a systematic review could identify the term “eating together” as being important for metabolic indices for nutritional health [3], even though the number of studies was too small for any sensible detailed analysis to be made.

For the nutritionally interested—data on commensality are almost never gathered in connection with large dietary surveys. In dietary assessment tool portals like the Diet, Anthropometry, Physical Activity (DAPA) Measurement Toolkit from the Medical Research Council in the UK, all different types of dietary assessment, subjective and objective methods, are listed for easy access. However, the only reference given to commensality is the following: “Individuals can record the time, location and whether they consumed meals alone or with others for each eating occasion, providing information on eating patterns and the eating environment” [4]. No additional information is available in this otherwise very well-equipped resource. Large studies such as NHANES in the United States [5] have not studied eating together and the latest Swedish dietary surveys [6,7] did not include any angle on commensality, just to mention a few. Commensality seems not to be assessed in dietary surveys performed on population level or representative parts thereof, such as different types of registrations or records of food intake, at least not using the word “commensality.” This means that the importance of commensality for a healthy diet cannot easily be studied from already existing data from dietary intake studies. The type of venue where food is eaten is more often recorded than whether eaten alone or accompanied by others.

In sociological or anthropological studies of eating situations, commensality is sometimes more thoroughly discussed and assessed [8]. In this regard, assessment of eating together can be studied for example using observations, interviews, or questionnaires.

A reasonable way forward, for the scholar with an interest in how eating together affects our wellbeing as well as what we eat, with whom, where, and how, would be to try to identify different ways of assessment of commensality. This could possibly lead to a broader understanding of the level of importance of eating together over research boundaries and lead to an increased comparability of studies [9].

The aim of this paper is to make a scoping review of quantitative as well as qualitative methods of studying commensality, in nutritional surveys as well as in surveys of other character, such as cultural, sociological, or anthropological studies. We also wanted to identify in which disciplines papers were published in the area as well as how the number of papers using the term commensality has evolved over time.

## 2. Materials and Methods

A scoping review mapping the body of a specific topic was used to summarize and disseminate relevant literature [10]. This scoping review followed the steps suggested by Arksey and O’Malley [11]: (1) identifying the research question, (2) identifying relevant literature, (3) selecting the studies, (4) charting the data and (5) collating, summarizing, and reporting the results. The research questions were: How has commensality been assessed in research studies? Which disciplines and which affiliations do researchers have who have published papers using the term commensality? Finally, how has the number of papers evolved over time?

### Literature Search, Database, and Search Terms

The database searched was Web of Science Core Collection, up to April 2019. Search term used: commensality. The following criteria had to be met for a study to be included:Published up to April 2019.Published in English.Somehow assessing commensalityDescribing commensality among humans.

We included 61 studies, whereof most used qualitative research methods and some were of quantitative character, including very few dietary surveys.

In the following, we have gone through the quantitative as well as the qualitative papers in some detail, and we produced a number of tables including all papers, listed under six categories.

## 3. Results

The papers we identified were roughly categorized into quantitative and different categories of qualitative papers, such as observations, interviews, and field studies.

### 3.1. Quantitative Research Studies

Nine papers (Table 1) [12,13,14,15,16,17,18,19,20] were identified, which used quantitative approaches of some sort to assess commensality, sometimes in combination with qualitative methods and common dietary survey methodology. One paper described a new questionnaire, Well-Being related to Food Questionnaire (Well-BFQ), which is waiting to be validated [14]. Most of the studies using questionnaires were cross-sectional studies [12,13,15,16,17,18,19], some of them performed at two different time points, describing development over time [12,15]. One study looked at only breakfast habits [13]. Two of the papers described the same data [15,16]. Finally, two studies, described in three papers, were using a common dietary survey technique (24-h recall) to collect high-quality data of actual intake rather than “usual” intake [15,16,19].

A common problem in most studies was that the response rates were low and in some cases clearly skewed, with non-responders of lower education level. Some suggestions that could be read out from the scoping review regarding the use of quantitative methods to assess commensality were recommending the use of mixed methods approaches, applying rigorous sampling techniques, including type of venue in the investigation (lowbrow, highbrow) and whether the venue facilitated commensality or not. Mailed or online questionnaires were used in several studies. Many authors suggested inclusion of with whom, where, how, when, questions. In all studies using questionnaires, socio-demographic characteristics were collected, in a more or less rigorous fashion and of course, ethical approval was sought, with informed consent collected from participants as soon as issues of sensitive nature were collected.

### 3.2. Qualitative Research Studies

Qualitative studies are described as those involving the systematic collection, organization, and interpretation of textual material derived from talk or observation [9]. Qualitative studies are used in the exploration of meanings of social phenomena as experienced by individuals themselves, in their natural context [21]. Among the identified papers on commensality assessment, 51 papers were categorized as qualitative. These were put into subcategories, such as qualitative interviews (Table 2) [22,23,24,25,26,27,28,29,30,31,32,33,34,35,36,37,38,39,40,41] and ethnological studies (Table 3) [42,43,44,45,46,47], log book plus interview (Table 4) [48,49,50,51] case studies (Table 5) [52,53] and diverse methods (Table 6) [54,55,56,57,58,59,60,61,62,63,64,65,66,67,68,69,70,71,72,73].

Among these qualitative papers, two approaches were identified. The first one studying commensality to understand a phenomenon within meals, and the second to examine how to understand commensality from a meal point of view. Several papers revealed aspects of different social meanings attached to food and commensality, as well as how meaning seems to have changed over time [27,36,51]. There was also some variations around the concept, such as; commensal space, meal, eating practice, conviviality, convivial dining [18,45,68].

Moreover, the diversity of the applicability of the method differed within qualitative methods, ranging from more or less rather standardized questionnaires [14,26], with specific open questions, to collections of life stories [27,34]. We could also see the same variation in how data have been analyzed ranging from statistical analysis such as regression analysis [50] content analysis [65] or using grounded theory [35,41].

According to different application of theories, theories surrounding “practices” were observed as the most commonly used in interview studies [24,27,30,34], within ethnographic methodology [31] as well as in questionnaires [26]. As commensality was seen as an important practice that appears in different settings and contexts, we thereby identified a perception of the importance of commensality research for more theoretical approaches on meal and food studies.

One of the most common themes on the qualitative papers was in regards to family eating [35,37,44,48,56,57,62]. None of these papers investigated how to implement commensality in research, but still discussed how to understand commensality in the use of technology during meals, family rites, parents/mothers and, family activities and class. All papers emphasized the importance of the use of commensality in research as a notion for bringing new light to the understanding of meals. Another interesting finding in the analysis of the papers was in regards to pleasure and health aspects of eating together. Here in the qualitative part of our results, the identified commensality papers seemed to concentrate on enjoyment or pleasure of eating together. This was most obvious in Phull et al. [45], discussing that overall economic, time and social pressures may inhibit pleasurable family dining.

### 3.3. Disciplines Identified in This Scoping Review and Number of Papers over Time

The final papers showing up in our review turned out to be mostly from the nutrition area (28 papers); 20 of them were published in Appetite. Clinical or medical journals including pediatrics published another large part (6 papers), social science (20 papers), ethnology, anthropology (6 papers), tourism studies (1 paper), urban studies (1 paper). It seems as if the search term commensality was used in several disciplines. We did not put a limit on how old the paper in WoS could be, yet the first paper using the term commensality and fitting the inclusion criteria was identified in the year 2000. From 2000–2014, the number of papers was fairly low, only ten papers were found during this fourteen year time span. In 2015, 7 papers were found, 2016 14 papers, 2017 17 papers, 2018 10 papers and 2019 up to April 4 papers all including some type of description of assessment of commensality. Countries involved in commensality studies according to our search were mostly European, with some originating from New Zealand, Japan, Korea, African countries, USA, Canada and South America. Sometimes authors studied other cultures than their own.

## 4. Discussion

The diversity of disciplines identified in this review is broad and the methods used to describe or to assess commensality show the true multidisciplinarity of the subject. An important result is also that the term commensality as search term seems to be widely spread over disciplines over the last five years, while earlier research is scarce. A large proportion of the commensality research identified in this review was European, possibly indicating that the term commensality has been more used in the European arena.

A great deal of the identified research is focused on health and nutrition, aiming to investigate the importance of commensality for health and wellbeing [12,13,14,15,16,18,20,22,24,26,30,33,39,46,58,65,67,69]. This means that data on health and health related behaviors, living conditions and sociodemographic data, need to accompany the data on commensality. To gather valid data on dietary intake, solid dietary survey methodology should be used. 24-h recall in questionnaire format, with at least 2 24-h recalls per individual is suggested as a good way to assess dietary quality [4,12,15,16,19]. Since 24-h recall is a type of diary, it can easily be connected to questions on with whom, where, type of venue, time of day, and type of meal consumed (breakfast, lunch, dinner, other). Furthermore, several nutritionally oriented authors expressed the need for including qualitative measures to complement the traditional quantitative data gathering [15,16,19], such as open-ended questions or combining questionnaires/diaries with interviews or observations for example.

There are a couple of questionnaires that could be used for authors only interested in commensality as such [18,19], while one is under development [14]. These questionnaires seem very comprehensive and describe wellbeing related to food, with commensality as a small part. Several authors underline the importance of employing proper sampling methods and techniques to reduce non-response or at least to try to identify a sample that finally provides you with a representative proportion of the population under investigation.

It is very surprising that so few of regular dietary surveys that are undertaken actually look at commensality, venue or eating alone [13,16,17]. Considering how often a stressful working life hinders commensality at the work place or at home, it is also important to assess time and place for working meals, which we decided to address in a separate paper.

The identified qualitative studies showed that the notion of commensality to a great deal had raised interest among researchers studying meals as a central part of health or wellbeing [23,24,25,26,27,28,31,33,36,37,39,41,45,50,54,55,65,67]. Meal studies are an important part of the analysis of food and eating [74,75], as meals contribute to social life as well as impact individual behavior [8,76]. This was shown by the identification of different norms regarding how certain meals ought to be shared and eaten together, using different qualitative research approaches. We identified suggestions for longitudinal study designs for cohorts over an extended period of time, in order to provide a higher degree of understanding of, for example routinization of eating behavior in regards to commensality. Furthermore, to reach a broader understanding of commensality as such over research boundaries and an increased comparability of studies, a need was identified to find ways to agree on how to interpret research findings, and make them compatible especially between qualitative and quantitative methodology, as suggested by Malterud [9]. This by applying terms such as reflexivity and transferability applicable for both qualitative and quantitative methodology. This would also gain further possibility for inter/multidisciplinary studies, improving our understanding of commensal meals for wellbeing as well as in cultural/social studies. As in Phull et al. [45], discussing that economic, time and social pressures may inhibit pleasurable family dining, we see an importance of combining qualitative results with quantitative “dependent variables” for further investigating the relations between healthy and pleasurable eating patterns, as suggested by Scander [77], as a gap between “good taste” and “good health” were identified. This was partly observed by interdisciplinary studies between Nutrition, Sociology and Culinary Arts and Meal Science, using both quantitative and qualitative methodology.

We could not identify any particular differences in assessment methods between western and eastern studies. Differences in methodology between studies of elderly and the general population could not be established.

### Strengths and Limitations and Suggestions for Future Research

This scoping review used only one database (Web of Science Core Collection) and did not include other search terms than commensality. It could certainly have been made more comprehensive by using other search terms such as eating together, family meal, or other terms and using several databases. The concept of commensality seemed to have been more used over time, especially during the last five years. This could also mean that we missed some early studies performed with a different choice of words for commensal eating, such as “family meals,” “eating together,” or “social eating”. A full systematic review should include more complete search terms and other databases [10,11]. However, it seems clear from previous systematic reviews that the definitions for these terms are often vague [2] and that the area of research is not sufficiently studied to be able to identify and disentangle relationships between eating together and metabolic indices of health [3]. We found it important to further explore aspects of timing when eating took place, why we extracted a small number of papers that concentrated on this topic and wrote a more in-depth paper on timing.

The strengths were that the high-quality database Web of Science Core Collection was used, which only includes papers published in peer-reviewed journals. The results of the search were of a magnitude that made it possible to review in a relatively rapid manner and the papers identified represented widely different disciplines. We wanted to highlight the multidisciplinary character of commensality and the included papers point at the developing field of commensality research, with commonalities and diversities in research methods. Using a quick and easy method of a scoping review provided us with a good overview of the papers using the specific term “commensality,” as a part of a project collection on diet, nutrition, and health with a focus on eating together [78]. Learning from each other on how to perform high-quality research on commensality has become even more important in this era of COVID-19, when commensality is challenged.

## 5. Conclusions

We conclude that in order to gather information on the importance of commensality for health and wellbeing, we should combine valid dietary survey methodology with valid estimates of commensality. Adding simple questions on where and with whom should be easy to integrate to large-scale dietary surveys. For all studies of commensality, a mix of qualitative and quantitative methods is recommended. Use of the term commensality in research papers should be encouraged, as well as providing more solid definitions of commensality and other search terms of eating together.

## Figures and Tables

**Table 1 ijerph-18-02632-t001:** Quantitative studies: Summary of the reviewed studies (n = 9).

Authors (Year), Country	Objectives	Methods, Settings and Participants	Results/Discussions	Results in Relation to Review
De Backer, CJS, (2013) [12] Belgium	To investigate if reported childhood food habits predict the food habits of students.	Cross-sectional survey of 104 higher education students in Belgium. Convenience sample.	Students appeared to maintain recalled childhood food rituals, mainly the matrilineal dominance. Breakfast and dinner patterns especially influenced commensality.	Frequencies of recalled childhood family meals influenced the frequencies of current commensality. More so for breakfasts and dinners than for lunches.
Gotthelf SJ, Tempestti CP. (2017) [13] Argentina	To investigate relationship between breakfast, sociodemographic outcome measures, and nutritional status among school children.	283 children 9–13 years of age, attending schools in the City of Salta, urban and peri-urban areas. Questionnaires, cross-sectional study.	Children of lower educated parents were at higher risk of skipping breakfast, which also related to having breakfast alone.	Only breakfast habits assessed, related to nutritional status and socioeconomic status.
Guillemin I, et al. (2016) [14] France	To develop and validate an instrument for assessing well-being associated with food and eating in an adult population.	A preliminary validation was made with 444 subjects with balanced diet, non-balanced diet and standard diet. Factor analysis.	The final Well-BFQ was shown to be a unique, modular tool that assesses the full picture of well-being related to food and eating habits.	The questionnaire is not available in English as yet, and a bigger validation study is underway.
Holm, L et al. (2016) [15] Danmark	To analyze changes in the social organization of eating over time.	Two cross-sectional surveys; in four Nordic countries 1997–2012: 1997 4808, 2012 8248 individuals 15 and older, 24 h recall of diet.	Some differences were seen between 1997 and 2012, not consistent between countries. More solitary meals and quick meals could be traced over the years. Single households stood out.	The authors strongly recommend the use of a mixed-methods methodology with 24-h recall.
Lund TB, er al. (2017) [16] Danmark	To analyze eating out in the Nordic countries in relation to work or leisure activity, related to some factors in people’s lives.	8248 individuals 15–80 years, Internet-based survey. The response rate was low, 9–11%. The lowest educated segment was underrepresented.	Occasional eating out in the Nordic countries. More people ate out in Sweden and Finland compared to the other countries. Often linked to work-life in urban settings.	Suggests employing measures that distinguish between eating venues (lowbrow vs highbrow) and kind (restaurant, café), the money spent on eating out and records of physical features of the venue that facilitate communication.
Marquis M, et al. (2018) [17] Canada	To explore environmental, personal, and behavioral factors as determinants of food behavior.	857 university students in Quebec, Canada ate, 20% alone in apt, 11% shared with a partner and 5% lived with parents	Eater profiles developed from factor analysis: the planet-nutrition-kitchen lover, the utilitarian loneleater, the body-driven eater, and the mindless eater.	Fairly simple questionnaire based on a number of published statements. These were analyzed in relation to gender and place of residence. The authors suggest further studies of food insecurity, health, and differences in resilience.
Sobal J, Nelson MK (2003) [18] USA	To investigate usual meal partners in commensal units and frequency of eating with others.	663 adults in a US community responding to a mailed questionnaire.	A useful graphic presentation of eating meals alone and eating with others: 27% ate no meals alone while 72% rarely ate with others weekly.	Most questions used are available in the paper. A substantial amount of eating is done alone, more than half ate breakfast alone but few ate dinners by themselves.
Yates L, Warde A. (2017) [19] Great Britain	Investigating aspects of British meal patterns, provisioning and preparation, timings and commensality.	Sample from a supermarket consumer panel. 2784 individuals, 45% response rate. Older, more affluent, better educated respondents and respondents without children were overrepresented.	Household members are the most common companions at meals (75%), work colleagues (16%). Meals later in the day are more probably eaten in companionship. When singles eat alone they eat simpler dishes.	The sampling is of utmost importance for the result. This type of survey can be performed to investigate commensality in a comprehensive manner, and in agreement with suggestions of Holm et al. Adult-only households were underrepresented in this sample. Timing was included in the analysis.
Sato et al. (2015) [20] Brazil	Investigated family meals and explored associations with family and sociodemographic characteristics, BMI, and eating practices	A population-based cross-sectional study, using complex cluster-sampling, conducted in the city of Santos, Brazil with 439 mothers.	Family meals were 54% more prevalent among mothers with high compared to low education. Eating no meals with family was more prevalent among those reporting that eating was one of the biggest pleasures.	The authors suggest the need for further research investigating the effects of family meals on mothers’ health through nutritional and phenomenological approaches.

**Table 2 ijerph-18-02632-t002:** Interviews: summary of the reviewed studies (n = 20).

Authors (Year), Country	Objectives	Methods, Settings and Participants	Results/Discussions	Results in Relation to Review
Andersen et al. (2015) [22] Denmark	To broaden our understanding of the concept of commensality by investigating what it means to “share a meal.”	Study: a hot meal based on Nordic ingredients vs the normal Danish school meal arrangement in which children bring lunch packs. In-depth interviews with teachers, chefs and staff, focus group interviews with pupils.	The study showed how different types of school meal arrangement influenced the social life of a school class, and how these arrangements involved strategies of inclusion and exclusion.	The results fail to confirm the conventional view that shared meals have greater social impacts and benefits than eating individualized foods. The article argues that the social entrepreneurship involved in sharing individual lunch packs might even outweigh some of the benefits of shared meals where everyone is served the same food.
Backett-Milburn (2010) [23] UK	To understand more about the social and cultural conditions which might be promoting more positive dietary health and physical well-being amongst middle class families	Parents/main food providers of boys and girls aged 13/14 years Eastern Scotland, Qualitative interviews in parents’ homes. Topic guide.	Most parents’ accounts appeared rooted in a taken-for-grantedness that family members enjoyed good health, lived in secure and unthreatening environments regarding health and resources, and able to lead active lives.	Parents described attempts to achieve family eating practices such as commensality, cooking from scratch, and encouraging a varied and nutritional “adult” diet and cosmopolitan tastes, but work and activities could compromize these
Bailey (2017) [24] Netherlands	To examine how the travel of food, food practices, and commensality reflect the flow of norms, practices, identities, and social capital between India and the Netherlands.	30 in-depth interviews conducted among Indian migrants living in The Netherlands	The main themes from the data included food from home, cooking practices, food sharing, and family relationships. Migrants’ sense of belonging was related to the food they brought from home and the memories it generated.	Commensality with co-ethnics led to a sense of community and stronger community bonds. Commensality with other non-Indian groups was perceived to be problematic. The exchanges of food, eating practices, and care created a sense of “co-presence” in lives of migrants.
Belon (2016) [25] Canada	To identify the barriers to and opportunities for healthy eating among residents of four communities representing the heterogeneity of urban communities.	A total of 35 individuals participated, from four communities in the province of Alberta, representing a spectrum of urban communities as defined by Statistics Canada, semi-structured interviews one-on-one were used.	This study identifies a set of meta-themes that summarize and illustrate the interrelationships between environmental attributes, people’s perceptions, and eating behaviors	This paper recognizes interrelationships among multiple environmental factors that may help efforts to design effective community-based interventions and address knowledge gaps on how sociocultural, economic, and political environments intersect with physical worlds
Cho (2015) [26] Korea	To examine cross-cultural variations of perceptions and actual practices of commensality and solo-eating.	University students in urban Korea and Japan, survey and self-administered questionnaire.	More Korean students reported they prefer commensality and tend to eat more when they eat commensally. Japanese reported no preference on commensality and there was no notable difference in food quantities.	The study revealed cross-cultural variations of perceptions and practices of commensality and solo-eating in non-western settings. Open ended question questionnaire.
Danesi (2018) [27] Switzerland	To contribute to research on social and cultural values of commensality and on the contemporary debate on changes of eating patterns by considering European young people’s food sharing practices	French, German, and Spanish young adult, In-depth semi-structured interviews and observations	The different nationalities contribute to emphasizing cultural diversity and laying the foundations for highlighting poignant differences between countries, in relation to meal times, content, places of food sociability, social organization at shared meals and the role of food sharing.	These aspects reveal different social meanings attached to food and commensality, as well as variability of commensal forms between young people living in or coming from different European countries
Dodds, Chamberlain (2016) [28] New Zealand	To analyze the content of a weekly nutrition column in a popular New Zealand magazine, from a social constructionist perspective.	New Zealand magazine text analyses	The articles advocated eating for health, but depicted nutritional information as open to interpretation. Fear-based messages were used by linking “unhealthy” food choices with fatness and chronic ill health.	Unhealthy foods were portrayed as more enjoyable than healthy foods, social occasions involving food were constructed as problematic.
Fossgard (2018) [29] Norway	To study how students experience and perceive their packed lunches and lunch breaks and to what extent the lunch break is a space for children’s sociality and for teachers’ governmentality?	11-year-old Students, 165 participants, focus-group discussions.	Students expressed that they appreciated their own packed lunches since they could decide what to eat. Shortage of time and disturbance in the classroom could ruin a good meal. The main issue raised was with whom they could sit and eat their packed lunches.	Findings underline the importance of considering the emotional dimensions of eating and that commensal eating is not dependent on sharing the same food. The children experienced that the lunch break was governed by an adult agenda in which they had limited opportunities to create their own spaces.
Giacoman (2016) [30] Chile	To examine the significance of communal eating among adults from Santiago, Chile.	24 group interviews were conducted in Santiago with family members, coworkers, and friends who shared meals with one another.	The results showed that the practice of commensality strengthens the cohesion among the members of a group. However, eating together also is assigned an ambiguous value	On the one hand, commensality is viewed as positive in enabling connections with others. On the other hand, participating in commensality can be viewed as negative, depending on the characteristics of the commensal group and the context, something that also was revealed by this study.
Neely (2014) [31] New Zealand	To explore the promotion of school connectedness through the practice of shared lunches within a secondary school.	Teachers and 16–18-year-old students in a New Zealand secondary school, ethnographic interviews	Shared lunches fostered common humanity, created an informal setting, encouraging sharing, enabling inclusive participation, demonstrating sacrifice for the communal good, and facilitating experiences of diversity.	Shared lunches, as part of an overall strategy to develop a well-connected school community, are adaptable and can fit into a multitude of situations to meet different needs. The findings of this study contributed to understanding the mechanisms by which shared lunches can affect indicators of school connectedness.
Neuman et al. [32] (2017) Sweden	To explore how 31 Swedish men talk about the sociality of domestic cooking in everyday life	31 Swedish men 22–88 years old, interviews with interview guide	Domestic cooking studies can show how Swedish men express sociality of cooking, intertwined with accomplishments of masculinity. The sociality of cooking is a way for men to maintain heterosocial relationships and assume domestic responsibility.	The paper discusses a potential cultural transition in men’s domestic meal sociality and suggest the need for studies to analyze how cooking shares similar properties to commensality, and the implications of this regarding gender relations.
Nyberg, Lennernäs Wiklund (2016) [33] Sweden	To investigate how the organization of work, time, and place influence the food and meal situation, focusing on patterns, form and social context of meals.	Flight attendants (Scandinavia), qualitative semi-structured interviews.	The organization of work, time, and place had a major influence on the meal situation and how meals were managed by FAs. The work was fragmented and inconsistent resulting in scattered meals and a more snack-based form of eating.	The findings demonstrated the individual responsibility to solve the meal at work, e.g., to solve organizational times
Scagliusi (2016) [34] Brazil	To analyze working mothers’ discourses about family meals eaten at the table, on the couch, and in the bed/bedroom.	30 mothers working in public universities of a Brazilian region Semi-structured interviews.	The table is a significant component of meals. Regarding the couch, it seems that the family chose to eat there, as a more casual and relaxed setting. Eating in bed was related to precarity, intimacy, and casualness. In the three settings, watching television was a common practice.	Commensality was seen as an important practice in different settings and contexts. The table emerged as the cornerstone of commensality. When a table was not present, new arrangements were made. Especially the couch seems to be a new commensal space, able to allow some collective conviviality. Finally, the significant role that television assumed in meals was also highlighted.
Schänzel, Lynch (2015) [35] New Zealand	To understand the individual and collective experiences and meanings of family holidays over time.	Ten New Zealand families consisting of 10 fathers, 10 mothers, and 20 children interviewed in homes.	Positive and negative memories of hospitality encounters for different family members are illustrated through the emotive concepts of commensality and spatiality.	Family meals take on symbolic and publicly celebrated characteristics, whereas shared accommodation space is contested. Theoretical implications of the nature of family hospitality dimensions are further discussed.
Sidenvall et al. (2000) [36] Sweden	To delineate the meaning of preparing, cooking, and serving meals among retired single living cohabiting women.	Sixty-three women living in two Swedish cities and their rural surroundings participated in qualitative interviews, home visits.	The whole procedure of preparing a meal could be seen as preparing a gift. Four phases were identified: finding out what to serve, cooking, presenting the gift in a beautiful manner, and enjoying the commensality.	Cohabiting women went on cooking with duty and joy as they had done before retirement. For widows, especially those who had recently lost their spouse, the meaning of cooking and eating was lost, and among these women there was a risk of poor nutritional intake.
Skafida (2013) [37] UK	To explore the extent to which family meal occurrence, meal patterns and perceived meal enjoyment predict the quality of children’s diets.	Scottish sample of five-year-old children, face-to-face structured interviews with child and mother.	Eating the same food as parents is the aspect of family meals most strongly linked to better diets in children, highlighting the detrimental effect in the rise of “children’s food.”	The results suggested that eating together was a far less important aspect of family meals and redirects attention away from issues of form and function towards issues of food choice. Policy implications and the importance for public health to recognize the way eating habits are defined and reproduce social and cultural capital are discussed.
Sobal, Bove Rauschenbach (2002) [38] USA	To study commensal patterns of people entering marriage.	Twenty couples in the USA, in depth semi-structured interviews.	Meal commensality varied across the daily cycle: Many spouses skipped breakfast or ate breakfast separately, most ate lunch at work, and dinner was the main commensal meal.	Greater marital commensality occurred on weekends than weekdays. Kin were major participants in commensal circles, with friends, coworkers, and neighbors also included as eating partners.
Tessler S, Gerber M. (2005) [39] France	To evaluate food habits in a holistic way, identifying key elements of the Mediterranean dietary model.	Mother-daughter couples in Sardinia (63) and Malta (61), qualitative questionnaire, open-ended, anthropological study.	The evening meal was seen to be the socializing meal in both islands. In Sardinia, both lunch and dinner were eaten together with family at home.	Mostly open-ended questions, which means a qualitative approach, showing the importance of the evening meal for socialization, lunches involving family members.
Traphagan JW, Brown LK. (2019) [40] Japan	To study eating patterns and attitudes within fast food restaurants in Japan.	Previously collected, ethnographic free listing, casual conversation interviews and observations, qualitative data from three cities in Japan.	The authors are claiming that McDonalds and other fast food chains are thriving in Japan due to the way such fast-food establishments resonate with other parts of Japanese life and culture.	Several different methods of collecting data, including free listing, casual conversation interviews and observations describing eating patterns and attitudes at fast food establishments in Japan
Vesnaver (2015) [41] GB	To explore experiences among older widowed women in relation to food behavior.	Interviews with 15 widowed women living alone in the community, aged 71 to 86 years.	Widowhood meant significantly fewer opportunities for commensality. Participants attributed changes to their food behaviors due to the experienced difference between shared meals and meals eaten alone, no longer having the commitment of commensality.	Free from the commitment of commensality, some shifted away from regular meals and simplified their meal preparation strategies. This has implications for clinical and research endeavors.

**Table 3 ijerph-18-02632-t003:** Ethnology: summary of the reviewed studies (n = 6).

Authors (Year), Country	Objectives	Method, Settings and Participants	Results/Discussions	Results in Relation to Review
Ahn Nelson (2015) [42] USA	To examine the behaviors and social interactions among preschool children and their teachers during food consumption at a daycare facility, using social cognitive theory.	Qualitative, ethnographic methods for studying pre-school children in a US daycare center.	Teachers’ food socialization styles and social interactions with peers cultivated children’s food consumption. Commensality rules set by the childcare institution also helped children learn other valuable behaviors.	The findings showed that teachers’ socialization styles and social interactions had a profound effect on children’s eating and meal time behaviors.
Benbow, H. M. (2018) [43] Australia	To study how sharing food created intercultural encounters that could be poignant and caring or alienating and divisive.	Australian soldiers and locals during the Battle of Timor in the Second World War. Memoirs and stories gathering	Discourse around food highlights the close connections formed between the commandos and their local “helpers,” the “criados.”	These food-related experiences were shown to have a particular poignancy to the enduring notion in the Australian military of a “Debt of Honour” owed to the people of Timor.
Osowski, C. P. and Y. M. Sydner (2019) [44] Sweden	To study children’s perceptions of meals with regard to what, where, and with whom meals are eaten and how meals are made.	112 Swedish children, ethnographic questionnaire.	Meals were often portrayed with family members sharing proper meals at home, spending enjoyable time together. The children described festive meals, which included extended family and friends, other foods and conditions.	The family meal functioned as a way to construct the family where children acquire norms and values about meals and family identity, in an active way, by breaking rules and by challenging norms. The commensal aspects surrounding included both commensal eating and commensal food work.
Phull, S., et al. (2015) [45] UK	To explore the concept of conviviality and its significance in relation to the Mediterranean Diet.	Review of lit to define conviviality.	Conviviality is widely used to promote the ideals of the Mediterranean Diet, Mediterranean culture, and family life. Definitions of the term conviviality suggest that convivial occasions are amicable, sociable occasions, and pleasure is experienced by all present.	It offers an interdisciplinary perspective on who and what makes conviviality happen and the potential obstacles to the experience and promotion of convivial dining.
Sabatini, F., et al. (2016). [46] Brazil	To analyze the experiences of nutrition students in constructing a portfolio about food and culture.	Twenty-nine students, blog following, family gatherings, a movie about food, analyzing a text, making a list of ideas, concepts and messages learnt. Eleven students also participated in focus groups	Important messages linking eating to history, culture, respect and commensality. In the focus groups, students highlighted that the portfolio helped them to reflect and to change their view of nutrition. The creation of the portfolio was considered exciting, involving and demanding	Students were stimulated toward a critical, humanized, and complex practice that respected the historical, social, cultural, sensorial, nutritional, and political nature of eating. The portfolio seemed a suitable learning method for complex themes, such as food and culture, providing a collaborative formation process
Woolley, K. and A. Fishbach (2017) [47] USA	To examine the consequences of incidental food consumption for trust and cooperation.	This study used a 2 (food: similar vs. dissimilar) vs 2 (role: investor vs. fund-manager) between-subjects design.	Food consumption further influences conflict resolution, with strangers who are assigned to eat similar foods cooperating more in a labor negotiation, and earn more money. Consumers are more trusting of information about non-food products. Food serves as a particularly strong cue of trust compared with incidental similarity.	Food serves as a particularly strong cue of trust compared with other incidental similarity. Theoretical and practical implications of this work for improving interactions between strangers, and for marketing products were discussed.

**Table 4 ijerph-18-02632-t004:** Log book plus interview: summary of the reviewed studies (n = 4).

Authors (Year), Country	Objectives	Methods, Settings and Participants	Results/Discussions	Results in Relation to Review
Jarosz (2017) [48] UK	To examine social differentiation in eating patterns in Britain.	Focus on family meals with under-aged children. Using data from the 2014–2015 UK Time Use Survey, 1 weekday and 1 weekend day.	The highest occupational class dedicated more time to family meals. This effect was no longer significant when controlling for education or income. Higher educated individuals had more frequent family meals, and more affluent individuals spent more time at the table with household members.	Household composition mattered for how people ate. Parents of younger children ate with them more frequently than parents of teenagers did. Single parents, a notoriously time-poor category, spent the least amount of time eating with their families.
Kniffin (2015) [49] USA	To investigate organizational benefits when coworkers engage in commensality.	Study performed within firehouses in a large city, mix of qualitative and quantitative including group conversations and interviews and introductory questions.	The field research showed a significant positive association between commensality and work-group performance.	The findings establish a basis for research and practice that focuses on ways that businesses or workplaces can enhance team performance by leveraging the mundane, powerful activity of eating together.
Kwon (2017) [50] Korea	To assess the association between eating alone and the MetS and to identify whether sociodemographic factors modify this association.	This study included 7725 adults and used interviews.	20.8% of men and 29.2% of women ate alone ≥2 times/day. Those who ate alone 2 or more times per day showed higher frequency of living alone, having no spouse, skip meals, and less eating out. Women with eating alone ≥2 times/day had a crude OR of 1.29 for MetS compared with women not eating alone.	Eating alone ≥2 times/day was significantly associated with increased abdominal obesity. This could be shown due to a combination of commensality and weight characteristics, interviews.
Paddock Warde Whillans (2017) [51] UK	This paper examines aspects of the experience of eating out in 2015 and its change over time.	A repeat survey, from quota sampling, conducted in the Spring of 2015. The third tranche of data arises from 31 follow-up, in-depth, semi-structured interviews conducted, in three cities.	Focus on the changing reasons and meanings of the activity as breadth of experience in the population augments and eating main meals outside the home becomes less exceptional	Ordinary events have become more prevalent, and the paper delineates two forms of “ordinary” occasions: the “impromptu” and the “regularized.” It describes the consequences for understanding the social significance of eating out, its informalization and normalization.

**Table 5 ijerph-18-02632-t005:** Case studies: Summary of the reviewed studies (n = 2).

Authors (Year), Country	Objectives	Method, Settings and Participants	Results/Discussions	Results in Relation to Review
Bardone (2017) [52] Estonia	To examine how pop-up restaurants challenge borders between private and public, business and entertainment.	Autoethnography and applied cultural analysis. Interviews, observations, website, Facebook, and weblog analysis including text and photos in the printed press	Culture theory as a set of complementary theories is seen to include and combine the approaches of different branches of science.	By extending the borders of the conventional restaurant or the home into public space, the pop-up restaurants create spaces of negotiation between the private and the public and new forms of commensality seemed to appear according to culture theory.
Marovelli (2018) [53] Ireland	This paper focuses on a type of commensality created by bringing together diverse participants beyond kinship relations and celebratory feasting.	Case study. Urban food sharing initiatives in London—a city which exhibits an active and dynamic urban food sharing ecosystem.	Social isolation and loneliness emerge as central drivers for participating in food sharing initiatives. Collective spaces and the affective qualities that they generate are particularly vital in urban contexts in times of austerity, as these initiatives have capacity to embrace social differences and to facilitate circulation of ideas and practices.	Identifies provisional bridging mechanisms between people, communities, projects, and services, exploring the connective ways which are hard to measure through quantitative measures and, are rarely articulated.

**Table 6 ijerph-18-02632-t006:** Diverse methods: summary of the reviewed studies (n = 21).

Authors (Year), Country	Objectives	Methods, Settings and Participants	Results/Discussions	Results in Relation to Review
Ares (2013) [54] Uruguay	The aim of the present work was to investigate consumers’ perception of wellbeing in a food-related context using an exploratory qualitative approach.	120 Uruguayan participants using three qualitative techniques: Word association, open-ended questions, and free listing.	The expected effects of foods on wellbeing were mainly related to non-communicable diseases such as high cholesterol levels, hypertension, and cardiovascular diseases.	Hedonic and emotional aspects of food consumption were important for consumers’ perceived wellbeing. Development of scales for measuring consumer perceived wellbeing when eating alone or in company.
Boulos (2016) [55] Lebanon	The purpose of the present study was to evaluate the association between three components of social isolation: social network, feeling of loneliness, commensality and nutritional status.	A total of 1200 randomly selected elderly individuals aged ≥65 years and living in rural areas of Lebanon participated in the present study. Face to face interviews.	Both social isolation and loneliness were independently associated with a higher risk of malnutrition. However, no association was found between the frequency of sharing meals and the risk of malnutrition.	The present study showed that social isolation and subjective loneliness are two independent risk factors for malnutrition among older people.
Ferdous (2016) [56] Australia	Exploring this tension and present a novel system TableTalk, which transforms personal devices into a communal shared display on the table to enrich mealtime interactions and experience.	Tabletalk design including nine families.	Our field study shows that TableTalk does not undermine togetherness, but supports familial expectations and experiences by stimulating conversation, reminiscing, bonding, education, and socializing.	The paper discusses how technology that is sensitive to the needs of family interactions can augment the commensal experience and reflect on design choices and opportunities that contribute, rather than disrupt, family mealtimes
Ferdous [57] (2017) Australia	This paper presents the deployment of a mealtime technology that orchestrates the sharing of personal devices and stories during family mealtimes, explores related content from all participants’ devices.	Seven families. Field study to use CHORUS. We began the initial visits at participants’ home with an interview with all members of the family including children.	Family interactions through sharing contents of personal and familial significance, supports togetherness and in-depth discussion by combining resources from multiple devices, helps to broach sensitive topics into familial conversation, and encourages participation from all family members.	This paper discusses the implications of this research and reflects on design choices and opportunities that can further enhance the family mealtime experience using the CHORUS mealtime technology.
Giacoman (2017) [58] Chile	The purpose of this paper is to explore the daily rhythms of eating, namely, the times at which food intake occurs during a day-long period.	The data used in this research come from a first time-use survey applied in Santiago in 2007 and 2008, which works with a retrospective activity journal to document the amount of time dedicated to different activities in 2282 cases during the 24 h of the previous day.	This study shows that people in Santiago tend to eat according to the same timetable (morning, midday and evening) and that socio-demographic variables have limited influence on the synchronization of this intake between Monday and Friday.	These results yield evidence that calls into question the applicability of the thesis of alimentary modernity within a Latin American context, which has not before been subject to investigation—time use study.
Gorman-Murray (2016) [59] Australia Canada	To examine the recent transformations in consumer landscapes and leisure spaces in inner city LGBT neighborhoods.	The study included consumer landscapes and leisure spaces in inner city LGBT neighborhoods in Sydney, Australia and Toronto, Canada, investigated through case studies.	Practices and spaces of leisure-based consumption are emerging in different forms across these neighborhoods. Deploying a discourse analysis of mainstream newspaper articles supplemented by relevant LGBT press releases, focusing on the use, meaning and social significance of leisure-based consumption sites—clubs, bars, cafes, restaurants.	The balance of daytime/night-time leisure spaces, which have both social and material affordances, is a key discriminator across the neighborhoods. Daytime consumer landscapes are often framed as sociable and inclusive within the media, while night-time landscapes are perceived as divisive.
Gorringe (2015) [60] USA	To present a qualitative descriptive case study of leadership and organization development philosophy and to summarize a strategy for using intentional organization design as a foundation for culturally aligned physician leadership development.	Mayo Clinic leadership development. Qualitative descriptive case study.	The authors identified four intentional characteristics of the multi-specialty group practice structure and culture that organically facilitate the development of leaders with the qualities required for the mission.	Development cannot be separated from the context and culture of organizational design. Organizational systems are designed to develop culturally aligned leaders, build social capital, grow employee engagement, foster collaboration, nurture collegiality, and engender trust. Organization design aligns the form and functions with leadership development and its mission.
Henig (2016) [61] UK	To explore how the notion of sofra [table/dining etiquette] is deployed by a Muslim Dervish brotherhood in a post- cosmopolitan town in post-war Bosnia-Herzegovina.	The aim was to show that mediations, orchestrated as hospitality events, can be conceived of as a form of sofra-diplomacy, in developing an ethnography of everyday diplomacy.	Suggests that the notion of sofra embodies both a mode of being diplomatic as well as a site of everyday diplomacy	The sofra enables the brotherhood to stage “events of hospitality” and mediate relationships between various “others,” locally and transnationally. There are two primary “sites” of post-cosmopolitan everyday diplomacy where shared urban civility, instantiated primarily through acts of commensality, namely the lodge and the soup kitchen.
Hopkins (2018) [62] Egypt	This paper describes the set of beliefs that underlie visit behavior and the sequence of events in visits to several of these shrines. Analysis was based on the human social relations.	The land in the Testour area in the early 1970s, historical anthropology. It is thus a form of historical anthropology. The approach is ethnographic, with a focus on people and events rather than on texts.	The visits involve prayer, commensality, and recreation. The shrines and the visits range from the simple to the complex. The occasional successful visit validates the system of beliefs and prepares for the next visit.	The collective aspect of the visits reinforces social relations within the groups of families and neighbors. These rituals can be transferred in the form of shared food and drink, and justifies a lot of commensality.
Lawson (2018) [63] UK	To study how exchange invited modes of collective, interconnected, and multifarious participation, which unsettled the cultural function of cakes as gendered, asymmetrical gifts, and the potential of cakes to develop generosity.	Fieldwork baking and investigating cakes in performance since 2009 through a series of works that invited the audience to share a memory of a significant person, place, time, or cake; in return, I baked their memories into unique cakes dedicated to those memories.	Through the signifying potential of cakes and the process of baking together, the performances evoked a multiplicity of complex emotional experiences. The baking performances echoed established symbolic uses of cake in commemorative practices including festive meals, celebrations, and religious ceremonies.	Cakes’ ability to move a community through the integration of commemoration, evidenced most acutely in the willingness of participants to embrace in the creative, symbolic potential of cake. Cakes are powerful objects in performance that can intervene in a community of strangers and map new avenues of commensality.
Madden (2015) [64] US	This essay to a special issue on utopian foodways acknowledges specific details of some of these changes and intersections, explains the origin of the issue.	Workshop at the University of Kansas, overview of included essays.	Thematically, they include analyses of images of abundance and lack in the past, “real” practices in the past and present, and imagined eating in the future.	The essay concludes with a call for “speculative” futures in research, writing, and teaching about utopian food practices including commensality.
Masson (2017) France [65]	To place dietary practices into a general context in which the relationship to food tends to be individualized and in which health-related issues remain an important aspect of the discourse about food on Internet.	Content analysis on fifty French individuals (25 females, 25 males) aged between 18 and 60 years old and with varied socio-economic profiles.	Food sharing is strongly promoted and solitary refection heavily frowned upon, the sole purpose of eating not being to nourish the physical body, but also to nourish the social body and connections with others.	The method, can be used for studying commensality. Although a personalized diet restricts the objective possibilities of food sharing, it is still central in representations of food, leads to the emergence of associated practices to introduce new forms of social eating behaviors, as those spread of the Internet.
O’Connor et al. (eds) 2015 [66] UK	To “analyze commensality in all its forms” and to make the point that “not all people in the world eat around a table or a common hearth—in fact, there are insurmountable variations of all kinds.”	This theoretically orientated edited volume proposes to examine sociality through six “mid-level concepts”: disjuncture, social field, social space, sociability, organization, and network.	“Commensality”—which at its most basic means “eating together”—suggests social relationships without defining or describing them in detail.	Unsurprisingly, there are many contesting definitions of commensality—the introduction lists them all, a long menu—leaving contributors free to choose their own interpretation, inevitably mixing levels of analysis. While food is a natural organizing theme, it does not play that role here.
Pachucki, et al. 2018 [67] USA	This study evaluates how social aspects of eating—frequencies of eating meals with others, meals prepared at home, and meals outside the home—are associated with nutrient intake.	Sex-stratified multiple regression analyses adjusted for confounders assessed the relationship between frequency of eating with others and nutrient intake multi-ethnic cohort type 2 diabetics re social aspects of diet	There was no consistently significant association between meals with others and the 5 nutrient intake measures for either men or women. The directions of association between categories of eating with others and diet quality—albeit not significant were different for men (positive) and women (mostly negative), which warrants further investigation.	Greater meal frequency at home was associated with significantly better scores on diet quality indices for men, while meal frequency outside the home was associated with poorer diet quality and energy intake for women. Better measurement of eating may inform ways to improve nutrition, especially for persons with diabetes for whom diet improvement can result in better disease outcomes.
Ray (2018) [68] USA	Aim was to paint an alternative picture of the streets taught to us by those on the move that forces us to re-imagine the national cultural space that has been allowed so far in considerations of good taste.	Streets and sidewalks are often read as mere targets of urban development schemes, rarely acknowledged for their importance to various forms of everyday life.	Western notions of public space can be seen as designed green places of relief from industrial and commercial concentration. Personal experience and a range of secondary literature an alternative way of thinking about the streets and the national cultural space is proposed, less officious, less bureaucratic, less national, than has been allowed so far in considerations of good taste.	Western notions of public space as designed green places of relief from urbanity are inadequate theoretical frames that cannot account for the sidewalk, the paan-shop, or the tea-stall, as sites of commensality, conviviality, and potential violence.
Rozin et al. 2006 [69] USA	To study attitudes toward large numbers of choices in the food domain.	Telephone interviews of 6000 representative adults from France, Germany, Italy, Switzerland, the UK, and the USA, included two items on attitudes to variety.	There was no substantial relation between a variety of demographic variables and variety preferences or expectations, except that older people were less inclined to prefer the high (50) variety in ice cream choices.	The results suggest that the US, and the UK to some extent, focus on providing choices that cater to individual differences in preferences, whereas the continental European countries are more attached to communal eating values.
Rozin et al. 2011 [70] USA	Three different questionnaires were distributed	Brief one-page (two-sided) questionnaires were distributed to individuals waiting alone in major train stations in Philadelphia and Paris. Questionnaire based on results from focus groups	Compared to the French, Americans emphasize quantity rather than quality in making choices, have a higher preference for variety, and usually prefer comforts over joys. American preference for quantity over quality is discussed in terms of the American focus on abundance as opposed to the French preference for moderation.	The American preference for variety is reflective of Americans’ more personal as opposed to communal food and other values.
Soulaimani (2017) [71] Morocco	Examines an oral story that displays how embodiment is crucial for understanding language in interaction.	A shopkeeper recounts an unusual dining event, started with commensality and ended with a physical dispute. Enacting its events, and displays stance through intricate forms of embodiment, and different ways of producing and copying gestures.	Analysis shows that telling a story, beyond a matter of verbal narrative, is performed through embodied enactments to allow for achieving co-participation and alignment	The data reveal a unique way of intensifying storytelling through both verbal and embodied repetition of selected story events.
Warde, Yates (2017) [72] UK	Understanding Eating Events: Snacks and Meal Patterns in Great Britain.	Conducted in Great Britain in 2012 (n = 2784), analyzed eating occasions which respondents stated were snacks rather than meals, focusing on their frequency, scheduling, contents, duration and social context.	Snacks take place in accordance with a common and predictable schedule. Snacks are smaller and less structured than most meals but mostly do not comprise what is generically described as “snack foods.” Snacks are shorter in duration, and less sociable than other eating events.	Most meals eaten in company, were planned for, and wholesome. Common negative evaluations between and outside meals have failed to appreciate the range of adjunct eating events and the degree to which they form part of aggregate patterns and individual routines. Interesting for snack events from commensality perspective.
Yount-André (2018) [73] France	This article examines how Senegalese immigrants, and their French-born children draw on eating practices to index religion as an axis of social differentiation, producing hierarchies of belonging in France.	120 ethnographic interviews with members of transnational Senegalese families, their kin, friends, and acquaintances in France and Senegal. Including observations, audio- and video-recorded meals and other acts of food sharing.	Even naturalized citizens feel pressure to permanently perform their integration according to the ever-shifting demands of French secularism, as “eating French” is increasingly defined in opposition to the practices of Muslim immigrants from France’s former colonies	French Republicanism contributes to a tiered form of citizenship through examination of the ways that educated migrants adopt the language of secularism to valorize their own eating practices relative to other transnational migrants.

## Data Availability

Data sharing not applicable.

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
