# Peer review of "Assessing Commensality in Research"

_ijerph, 2021, doi:10.3390/ijerph18052632_

Round 1

Reviewer 1 Report

Thank you for the opportunity to allow me to review the submitted manuscript. In the manuscript, the authors reviewed research conducted around "commensality" which is an important area of research that deserves more attention. 

Below are some minor suggestions for the authors to consider:

1. Could the authors clarify why terms such as "family meals" and "social eating" were not considered as a keyword for "commensality?"

For example, these several papers seem to be relatable to "commensality" yet not included in the submitted manuscript.

"Adolescent and parent views of family meals by Fulkerson et al", Breaking bread: the functions of social eating by Dunbar "

"Family meals and adolescents: what have we learned from Project EAT by Neumark-Sztainer et al"

"Family meals during adolescence are associated with higher diet quality and healthful meal patterns during young adulthood by Larson et al"

"The family meal: views of adolescents by Neumark-Sztainer et al"

"Women's social eating environment and its associations with dietary behavior and weight management by Motteli et al" 

2. Since the focus of this paper is not around COVID-19, I suggest considering removing parts where it integrates COVID-19.

3. I would consider removing line 89-91 as it distracts from the focus of this paper.

Author Response

Reviewer 1

  1. Could the authors clarify why terms such as "family meals" and "social eating" were not considered as a keyword for "commensality?"

For example, these several papers seem to be relatable to "commensality" yet not included in the submitted manuscript.

"Adolescent and parent views of family meals by Fulkerson et al", Breaking bread: the functions of social eating by Dunbar "

"Family meals and adolescents: what have we learned from Project EAT by Neumark-Sztainer et al"

"Family meals during adolescence are associated with higher diet quality and healthful meal patterns during young adulthood by Larson et al"

"The family meal: views of adolescents by Neumark-Sztainer et al"

"Women's social eating environment and its associations with dietary behavior and weight management by Motteli et al" 

Answer: Thank you for taking time to review our paper. The terms "family meals" and "social eating" were not considered as a keyword for "commensality" since we were interested in papers that used the exact term commensality. However, we added a few systematic reviews on family meals and social eating as a part of the introduction. Line 32-44, 261-265, 271-275

  1. Since the focus of this paper is not around COVID-19, I suggest considering removing parts where it integrates COVID-19.

Answer: Thank you for your comment. We actually asked all authors to our project collection to have some mentioning of Covid-19 aspects. But we have now moved it to the discussion and the part of future studies, mentioning it with one sentence only. Line 287-289

  1. I would consider removing line 89-91 as it distracts from the focus of this paper.

Answer:  Thank you, we understand, and point taken. we took that part out.

Reviewer 2 Report

Major comments

  The topic, commensality is important. However, this manuscript does not have formal analysis on commensality and just presented the list of ‘relevant’ papers. The methodology of selection is not valid. This does not stand as a scientific paper for the International Journal of Environmental Research and Public Health.

Minor comments

  1. Materials and Methods

Page 2, line 70: Please use a reference for scoping review mapping.

Page 2, line 71: “Which disciplines and which affiliations do researchers have who 75 have published papers using the term commensality?” I do not understand this sentence.

Page 2, line 79: April

Page 2, line 80: searching “commensality” is rather direct and narrow. I imagine that the authors overlooked many papers.

Page 2, line 87: Four inclusion criteria were met? Inclusion criteria are set in advance, and you choose studies which met the inclusion criteria.

Stopped here due to limited level of scientific writing.

Author Response

Reviewer 2

Major comments

  The topic, commensality is important. However, this manuscript does not have formal analysis on commensality and just presented the list of ‘relevant’ papers. The methodology of selection is not valid. This does not stand as a scientific paper for the International Journal of Environmental Research and Public Health.

Answer:  Thank you for commenting on our paper. Our attempt to reach a valid methodology is based on writing a Scoping review in line with Arksey O’Malley. We would like to point out the differences between performing a scoping review and a systematic review, and that we are aware of these. Therefore, we do not try to draw any such conclusions, but rather point out important observations in regards to our results.      

Minor comments

  1. Materials and Methods

Page 2, line 70: Please use a reference for scoping review mapping.

Answer:  Thank you, of course, we have now added reference on how we used scoping reviews and when and why scoping is relevant. Our mistake, the reference has been added, ref 11, as well as ref 10 referring to the “Guidance for authors when choosing between a systematic or scoping review”. Line 82-83

Munn, Z., et al., Systematic review or scoping review? Guidance for authors when choosing between a systematic or scoping review approach. BMC medical research methodology, 2018. 18(1): p. 1-7.

Hilary Arksey & Lisa O'Malley (2005) Scoping studies: towards a methodological framework, International Journal of Social Research Methodology, 8:1, 19-32, DOI: 10.1080/1364557032000119616

Page 2, line 71: “Which disciplines and which affiliations do researchers have who 75 have published papers using the term commensality?” I do not understand this sentence.

Answer:  We were looking for research disciplines and affiliations to understand in which research field “commensality” had been published. We have now changed the sentence to” In which disciplines and from which affiliations have the identified papers been published?”…. line 86

Page 2, line 79: April

Answer:  Thank you, changed to “April”. Line 85.

Page 2, line 80: searching “commensality” is rather direct and narrow. I imagine that the authors overlooked many papers.

Answer: Thank you for this comment, we wanted to identify papers using the word commensality to see if the term was more adopted over time, was used in certain disciplines or certain geographic areas. We have also emphasized the notions of ‘eating together’ and on ‘family meals’ in the introduction and discussion. Line 32-42, 270-274, 284-286

Page 2, line 87: Four inclusion criteria were met? Inclusion criteria are set in advance, and you choose studies which met the inclusion criteria.

Answer: Thank you for pointing this out. We have now changed the wording accordingly. “The following criteria had to be met for a study to be included”. Line 86.

Stopped here due to limited level of scientific writing.

Answer: We understand most of your review comments. We do hope that we met the criteria for a simple scoping review in line with Munn et al and Arksey o’Malley. (ref 10 and 11), and as a part of our project collection using a multi-disciplinary view of commensality, described in the Editorial paper (ref 78): Yngve, A., et al., The Project Collection Food, Nutrition and Health, with a Focus on Eating Together. International Journal of Environmental Research and Public Health, 2021. 18(4): p. 1572.

Reviewer 3 Report

The article "Assessing commensality in research" presents an interesting review that requires some important improvements to be accepted for publication in IJERPH.
- Table1 is too long, please remove redundant information or try to group the results so that the table is easier to read.
- Please include one or two figures (diagrams) that allow the reader to visualize improving the purpose and focus that the authors intend to print the article (one figure is proposed in the introduction and another in the discussion).
- In the discussion section, the differences found between the general population and the elderly should be highlighted. Furthermore, the possible differences between Western (developed) and Asian- emerging cultures are not mentioned.

Author Response

Reviewer 3

The article "Assessing commensality in research" presents an interesting review that requires some important improvements to be accepted for publication in IJERPH.

Answer:  Thank you for your review, we hope to be able to respond to your questions in a sensible manner.

- Table1 is too long, please remove redundant information or try to group the results so that the table is easier to read.

Answer: Thank you for this comment, we did freak out a bit when we produced the table! We have now worked on the table and discussed it with the editorial staff at IJERPH and hope that you now find it easier to read. Please note that the final layout of the table is done by the journal after the paper has been approved. We have now added lines between the papers to increase readability for reviewers.

- Please include one or two figures (diagrams) that allow the reader to visualize improving the purpose and focus that the authors intend to print the article (one figure is proposed in the introduction and another in the discussion).

Answer: We have tried to come up with suitable figures but found it difficult to do so due to the scoping nature of the work.

- In the discussion section, the differences found between the general population and the elderly should be highlighted. Furthermore, the possible differences between Western (developed) and Asian- emerging cultures are not mentioned.

Answer:  Thank you, we have now added a few sentences and hope that the following clarifies these issues.

Countries involved in commensality studies according to our search were mostly European, with some originating from New Zealand, Japan, Korea, African countries, USA, Canada and South America. Sometimes authors from Europe studied other cultures. Line 181-183

We could not identify any particular differences in assessment methods between western and eastern studies. Differences in methodology between studies of elderly and the general population could not be established. Line 261-283

Reviewer 4 Report

The manuscript “Assessing Comensality in Research” aimed to make a scoping review of quantitative as well as qualitative methods of studying commensality, in nutritional surveys as well as in surveys of other character, such as cultural, sociological or anthropological studies. In addition, the authors identify in which disciplines papers were published in the area as well as how the number of papers using the term commensality has evolved over time.

The methodology used was that of Arksey and O’Malley, 2015. This scoping review used only one database (Web of Science Core Collection) and did not include other search terms than commensality. The authors well justified this limitation so, it does not seem to be an impediment to the manuscript publication.

The manuscript is easy to read, and it is well written. The methodology as well as the critical analysis is well structured. However, please consider the following comments prior to final acceptance for publication in IJERPH:

  1. The reference to the method is missing in References

  1. Line 102 - put out the meaning of Well-BFQ

  1. Line 105 – remove de space “habits [8] .”

  1. I think the titles of the tables are clearer this way: “Quantitative studies: Summary of the reviewed studies (n = 9).” Check and correct if necessary

  1. In table 1, remove the line that appears in the middle of the table with: Authors (year), Country, Objectives…

  1. Tables need to be improved in order to make it easier to read and more appealing. The space between the different authors does not exist and it is sometimes very complicated to know, for example, where the objective of one author ends and that of another begins (see Table 2 the objectives of the authors “Backett-Milburn (2010) [17] UK ”and“ Bailey (2017) [18] Netherlands”.

  1. All tables must be reviewed and standardized, as there are phrases that end a period, others do not; some start with capital letters, others do not, some are justified, others are not; some "Authors (year), Country" and "Results in relation to review" are in bold others are not (Table 2), etc.

  1. Line 202 “measures to compliment the traditional” change to “measures to complement the traditional”

Author Response

Reviewer 4

The manuscript “Assessing Commensality in Research” aimed to make a scoping review of quantitative as well as qualitative methods of studying commensality, in nutritional surveys as well as in surveys of other character, such as cultural, sociological or anthropological studies. In addition, the authors identify in which disciplines papers were published in the area as well as how the number of papers using the term commensality has evolved over time.

The methodology used was that of Arksey and O’Malley, 2015. This scoping review used only one database (Web of Science Core Collection) and did not include other search terms than commensality. The authors well justified this limitation so, it does not seem to be an impediment to the manuscript publication.

The manuscript is easy to read, and it is well written. The methodology as well as the critical analysis is well structured. However, please consider the following comments prior to final acceptance for publication in IJERPH:

  1. The reference to the method is missing in References

Answer:  Thank you for pointing out the missing reference. Our mistake, the reference has been added, ref 11, as well as ref 10 refering to the “Guidance for authors when choosing between a systematic or scoping review”. Line 82-83

Munn, Z., et al., Systematic review or scoping review? Guidance for authors when choosing between a systematic or scoping review approach. BMC medical research methodology, 2018. 18(1): p. 1-7.

Hilary Arksey & Lisa O'Malley (2005) Scoping studies: towards a methodological framework, International Journal of Social Research Methodology, 8:1, 19-32, DOI: 10.1080/1364557032000119616

  1. Line 102 - put out the meaning of Well-BFQ

Answer:  Thank you, of course, we have now added the full meaning, “Well-Being related to Food Questionnaire (Well-BFQ)”. Line 109

  1. Line 105 – remove de space “habits [8] .”

Answer:  Space removed, thank you for noticing.

  1. I think the titles of the tables are clearer this way: “Quantitative studies: Summary of the reviewed studies (n = 9).” Check and correct if necessary

Answer: Thank you, we do agree to change order on table titles 1-6, as you propose.

  1. In table 1, remove the line that appears in the middle of the table with: Authors (year), Country, Objectives…

Answer:  Thank you, that has been corrected.

  1. Tables need to be improved in order to make it easier to read and more appealing. The space between the different authors does not exist and it is sometimes very complicated to know, for example, where the objective of one author ends and that of another begins (see Table 2 the objectives of the authors “Backett-Milburn (2010) [17] UK ”and“ Bailey (2017) [18] Netherlands”.

Answer:  Thank you we have now added extra space in between the reviewed papers. And in general, we have also discussed overall table layout with the editorial staff at IJERPH, to make it more readable. Please note that the final layout of the table is done by the journal after the paper has been approved. We have now added lines between the papers to increase readability for reviewers.

  1. All tables must be reviewed and standardized, as there are phrases that end a period, others do not; some start with capital letters, others do not, some are justified, others are not; some "Authors (year), Country" and "Results in relation to review" are in bold others are not (Table 2), etc.

Answer:  Yes, absolutely, we did freak out a bit when we produced the table, and we have now done an extended work on them. There were also some mishaps in the submission prosses, we have now in close discussion with editorial staff at IJERPH solved those problems.      

  1. Line 202 “measures to compliment the traditional” change to “measures to complement the traditional”

Answer:  Thank you for noticing, now changed to “complement”. Line 221.

Round 2

Reviewer 3 Report

Accept in present form

Reviewer 4 Report

The authors have significantly improved the manuscript, in my opinion, it is ready to be published.